# Hyaluronic Acid-Based Injective Medical Devices: In Vitro Characterization of Novel Formulations Containing Biofermentative Unsulfated Chondroitin or Extractive Sulfated One with Cyclodextrins

**DOI:** 10.3390/ph16101429

**Published:** 2023-10-09

**Authors:** Valentina Vassallo, Celeste Di Meo, Giuseppe Toro, Alberto Alfano, Giovanni Iolascon, Chiara Schiraldi

**Affiliations:** 1Department of Experimental Medicine, Section of Biotechnology, University of Campania “Luigi Vanvitelli”, 80138 Naples, Italy; valentina.vassallo@unicampania.it (V.V.); celeste.dimeo@unicampania.it (C.D.M.); alberto.alfano@unicampania.it (A.A.); 2Department of Medical and Surgical Specialties and Dentistry, University of Campania “Luigi Vanvitelli”, 80138 Naples, Italy; giuseppe.toro@unicampania.it (G.T.); giovanni.iolascon@unicampania.it (G.I.)

**Keywords:** hyaluronic acid, biofermentative chondroitin, articular cartilage, human primary pathological chondrocytes, osteoarthritis

## Abstract

Currently, chondroitin sulfate (CS) and hyaluronic acid (HA) pharma-grade forms are used for osteoarthritis (OA) management, CS as an oral formulations component, and HA as intra-articular injective medical devices. Recently, unsulfated chondroitin, obtained through biofermentative (BC) manufacturing, has been proposed for thermally stabilized injective preparation with HA. This study aimed to highlight the specific properties of two commercial injective medical devices, one based on HA/BC complexes and the other containing HA, extractive CS, and cyclodextrins, in order to provide valuable information for joint disease treatments. Their biophysical and biomechanical features were assayed; in addition, biological tests were performed on human pathological chondrocytes. Rheological measurements displayed similar behavior, with a slightly higher G′ for HA/BC, which also proved superior stability to the hyaluronidase attack. Both samples reduced the expression of specific OA-related biomarkers such as NF-kB, interleukin 6 (IL-6), and metalloprotease-13 (MMP-13). Moreover, HA/BC better ensured chondrocyte phenotype maintenance by up-regulating collagen type 2A1 (COLII) and aggrecan (AGN). Notwithstanding, the similarity of biomolecule components, the manufacturing process, raw materials characteristics, and specific concentration resulted in affecting the biomechanical and, more interestingly, the biochemical properties, suggesting potential better performances of HA/BC in joint disease treatment.

## 1. Introduction

Viscosupplementation was proposed a few decades ago for the management of (low-grade) osteoarthritis (OA) to postpone or reduce the pharmacological treatments, usually based on analgesics and non-steroidal anti-inflammatory drugs (NSAIDs) [1,2]. OA is the most common weakening disease in the global population, and its long-term symptoms can affect the patient’s life quality of life, causing joint aches and stiffness, fatigue, and sometimes, consequently, depression [3,4,5,6]. Additionally, pharmacological therapies, education, structured exercise, weight loss, and intra-articular injective treatments have gained importance in the last 20 years and have been exploited to treat joint injuries, such as the ones experienced by athletes [7,8]. In fact, local treatments may improve movement and reduce pain, eventually helping the regeneration/healing of the damaged tissues. As far as concerns viscoelasticity, hyaluronic acid (HA) is certainly the major player, being the highest molecular weight (MW) glycosaminoglycan (GAG), and thus the most effective in modulating viscoelastic properties of synovial fluid and extracellular matrix (ECM) biomechanical features [7,8].

Being a molecule with remarkable mechanical and biological properties, HA has countless applications [9]. In particular, its use as a functional biomaterial and drug delivery system is continuously widespread. In this context, an innovative 2-methylacryloyloxyethyl phosphorylcholine decorated methacrylate anhydride-hyaluronic acid drug delivery particle has been recently developed for OA treatment purposes [9,10]. As demonstrated by Yang and collaborators, this HA-based drug delivery particle was able to reduce the inflammatory process in both OA in vivo and in vitro models, opening new possibilities for therapeutic approaches [10]. Besides the latter scientific report, there are many others referring to chemically modified or crosslinked hyaluronan for intra-articular applications. Also, smaller molecules, such as cyclodextrin derivates (CDs), have relevant applications in food and pharmaceutical industries as nano-carriers [11].

However, most registered class III medical devices in this field still rely upon liner glycosaminoglycans (GAGs). It has been recognized that concentration and the MW of HA and other proteoglycans and/or GAGs affect the biochemical crosstalk within tissues that constitute joints [12,13,14]. In fact, HA and chondroitin sulfate (CS) are key components of articular cartilage, and it is well known that their content is diminished during OA inflammatory and degenerative process; thus, the assessment of treatments aimed at restoring their presence and/or increasing their concentration are needed [15]. In detail, CS also belongs to the GAGs family, it is ubiquitous in all vertebrates and invertebrates, and it is involved in several biological processes, such as maintenance of specific cellular phenotype, wound healing, and inhibition of pro-inflammatory cytokines activation and action [16]. This biopolymer comprises repeating disaccharide units of N-acetyl-D-galactosamine (GalNAc) and D-glucuronic acid (GlcA), and it is sulfated at the C-4 position or C-6 position. Additionally, the sulfation pattern and the chain length are variable depending on tissue age and sources [15,17]. Several scientific reports showed that sulfation patterns, source of extraction, processing and purification techniques, and molecular weight can affect the potential CS bioactivity [18,19]. For many decades, CS has been used as an active pharmaceutical ingredient in oral formulation, at very high purity and gram scale doses, or as a component in food supplements (FS) for OA management symptoms, with satisfactory results [20,21]. In fact, CS is able to inhibit the activation of NF-kB, the most important player in the inflammation progression, thus counteracting the expression of pro-inflammatory cytokines (e.g., IL-6 and TNF-α) and metalloproteases (MMPs) [22]. In this respect, a recent study showed that CS increased the expression of metallopeptidase inhibitor (TIMP) in an OA in vitro model [23]. In addition, CS plays a role in the maintenance of cartilage homeostasis, enhancing the expression of collagen type 2A1 (COLII) and aggrecan (AGN) and improving synovial fluid viscoelasticity [22,23]. Another fundamental contribution to the articular cartilage structure integrity is given by the cartilage oligomeric matrix protein (COMP), so its high expression at gene and protein levels by the articular cells is associated with damage and repair attempts [24]. In addition, the ability of CS to negatively modulate these biomarkers has been well confirmed in pathological human primary articular chondrocytes [25,26]. Recently, the intra-articular injection of HA combined with CS has been suggested to be effective in a small cohort of patients [27]. Rivera et al. conducted a prospective study on 112 patients with a grade 2 to 3 of knee OA, treated using three intra-articular injections of a novel device based on a combination of HA and CS [28]. The authors reported that the proposed intervention was able to assure pain relief and improved function over 6 months of follow-up [28]. Interestingly, the combined intra-articular injection of HA and CS was effective since the first application, probably due to an anti-inflammatory effect [29]. More interesting, a novel biofermentative unsulfated chondroitin has been obtained at a high purity grade by biotechnological inventive processes and reported a molecular weight like those of fish (shark fins) extractive chondroitin sulfate, this one being the larger molecules among extractive CS. Despite the presence of CS in the cartilage ECM and the synovia, it is very rarely present in injective treatments, probably because it has a MW of 20–40 kDa, 20–50-fold lower than the HA one, thus poorly affecting the rheology of water solutions [30]. This study aimed to explore the biophysical, mechanical, and biological effects of two different HA-based gels coupled either to marine CS or to unsulfated chondroitin already available in clinical practice. The unsulfated chondroitin, here used, has been obtained by a well-established fermentative process and specific purification procedures, and it is here named biofermentative chondroitin (BC) [31]. Previous studies on OA human synoviocytes, intra-proteome and secretome, highlighted a similar profile between CS and BC-treated cells but pointed out a major effectiveness of BC anti-inflammatory activity [32]. The hybrid complexes based on HHA and unsulfated chondroitin have been tested in a clinical study on symptomatic hip OA, demonstrating that they are tolerated, safe, and effective in improving movement and reducing pain [33]. Other comparisons between CS and BC have been performed regarding their ability to induce the differentiation of mesenchymal stem cells towards the chondrocyte phenotype both as additional components to the culture medium and as constituents of methacrylate gelatin-based biomaterials [34,35]. It is worth underlining that the potential introduction of BC in OA management symptoms may help in overcoming chondroitin extraction and purification-related problems as well as ethical (correlated to the use of animal sources) and religious issues [36]. In this experimental setup, based on human pathological chondrocytes, we aimed to compare in a full range characterization from the rheological behavior to stability features and biochemical/biological properties of CE-approved class III medical devices containing HA enriched with either CS or BC, containing a high final concentration of GAGs, namely 40 mg/mL which, to the best of our knowledge, represent the maximal titers available for injective treatments up to date.

## 2. Results

### 2.1. Rheological Characterization

The mechanical spectra of the samples were recorded and reported in Figure 1 for the preparations as distributed in the syringes. Both samples showed the typical viscoelasticity of unmodified hyaluronan-based formulation, with a viscous behavior at low frequencies (G″ exceeding G′) and an elastic behavior at high frequencies (G′ exceeding GG″). G′ increased more markedly than G″ until a crossover was observed in the range of 2–4 Hz.

The values of elastic and viscous moduli at 0.5 Hz and 2.5 Hz and the values of tan δ and complex viscosity η* corresponding to the same frequencies were extracted from the mechanical spectra and reported in Table 1, Table 2, and Table 3, respectively.

Storage and loss moduli resulted both slightly higher for the HHA/BC sample with respect to the ones obtained for HHA + CS + cd, while the values of tan δ seemed comparable between samples; data are consistent with the linear forms of the GAGs in solution. No significant differences could be found for the ratio GG″/G′ in all the range of frequencies of clinical interest. However, the formulation HHA/BC presented a complex viscosity higher than that of the HHA + CS + cd sample in the whole range of frequency considered; this behavior resembles the profiles of the dynamic viscosity as a function of shear rate. Particularly, the values of η* corresponding to 0.5 Hz and 2.5 Hz were statistically different (*p* < 0.05) between the two samples, while at the end of the shear thinning behavior, the difference is reduced and no longer evident (frequencies higher than 2.5 Hz).

### 2.2. Sample Enzymatic Degradation

Data on samples’ sensitivity to BTH-catalyzed hydrolysis are reported in Figure 2. Both samples showed a consistent molecular weight degradation during the incubation with the enzyme, with a more significant reduction for the sample HHA + CS + cd after 4 h of incubation. The residual weight fraction of molecules above 600 kDa was obtained: after the BTH attack, 47% of the MW fraction above 600 kDa was preserved for HHA/BC, while only 17% for HHA + CS + cd. A significant difference was found regarding the resistance to hyaluronidase digestion.

### 2.3. Cellular Viability Assay

The in vitro model proposed here is based on primary chondrocytes isolated from OA-affected patients, thus presenting an ongoing inflammatory/degradative process. In this way, the experimental setup may better resemble in vivo conditions. Although the two products tested here are registered in Europe, their effect in a pathological in vitro model, based on chondrocytes, is interesting to highlight eventually occurring effects due to contaminants even at very low concentrations (e.g., trace elements, metals, endotoxins, etc.). Therefore, our data that showed human pathological chondrocyte viability under treatments are relevant. In fact, Figure 3 shows that both HHA + CS + cd and HHA/BC sustained human chondrocyte viability within 72 h of treatment. In this context, HHA + CS + cd displayed a major effect after 48 h, then a plateau, increasing chondrocyte growth/viability by about 20% in comparison to pCTRL. The results proved that HHA/BC is more effective than HHA + CS + cd, prompting growth with a significant increase of cell viability of about 60% (* *p* < 0.05) after 72 h with respect to pCTRL. At this final experimental point time (72 h), HHA/BC viability support resulted as significantly (# *p* < 0.05) superior to HHA + CS + cd one.

### 2.4. Gene Expression Analyses via qRT-PCR

COLII and AGN are specific biomarkers of the chondrocyte phenotype and key components of ECM. Our outcomes showed that both HHA + CS + cd and HHA/BC were able to significantly (* *p* < 0.05) increase the gene expression of these biomarkers in comparison to pCTRL (Figure 4a). It is noteworthy that COLII was significantly increased at the gene level (# *p* < 0.05) in the presence of HHA/BC with respect to HHA + CS + cd. Specifically, COLII gene expression resulted in an up-regulation of about 2.8 and 5.4-fold in the presence of HHA + CS + cd and HHA/BC, respectively, while HHA + CS + cd and HHA/BC increased AGN gene expression by about 6.2 and 16-fold correspondingly and the effectiveness of both the medical devices resulted significant (*p* < 0.05) respect to untreated chondrocytes (pCTRL). Notably, both samples also lowered the expression of specific genes related to the inflammation process and cartilage degradation (Figure 4b) with respect to pCTRL. In fact, IL-6 and TNF-α were negatively modulated, confirming a strong anti-inflammatory effect of both products. However, HHA/BC appeared to have (# *p* < 0.05) more efficacy than HHA + CS + cd in affecting the IL-6 expression level. Additionally, for TNF-α down-regulation, the tested gels displayed a similar behavior. Interestingly, the expression of MMP-13, a degradative matrix enzyme involved in the OA cartilage degenerative process, also proved to be significantly (* *p* < 0.05) decreased by both HHA + CS + cd and HHA/BC (Figure 4b).

### 2.5. Protein Analyses via Western Blotting

Specific OA-related expression proteins were evaluated on pathological chondrocytes to evaluate the effects of HA coupled to sulfated or unsulfated chondroitin-based gels. In this context, COMP-2 and NF-kB were considered biomarkers connected to cartilage damage and inflammation. As expected, the untreated cells (pCTRL) expressed a marked protein level of the latter, confirming an ongoing cartilage degradation/damage and severe inflammation process. Our outcomes showed that HHA/BC had a significant (# *p* < 0.05) superior effect in COMP-2 protein expression reduction with respect to HHA + CS + cd (Figure 5a). However, in comparison to pCTRL, the reduction of this biomarker was about 1.9-fold (* *p* < 0.05) in the presence of HHA/BC and 1.3-fold with HHA + CS + cd. As shown in Figure 5a, both samples significantly (* *p* < 0.05) down-regulated NF-kB expression levels in comparison to untreated cells. In addition, biomarkers related to ECM degradation/remodeling were also selected. Specifically, the ability of HHA + CS + cd and HHA/BC to affect the protein expression of MMP-13 and TIMP-2 was assayed (Figure 5b). The analyses showed that both chondroitin-based treatments reduced MMP-13 protein levels, but HHA/BC was more effective than HHA + CS + cd with a fold reduction of about 1.7-fold vs. pCTRL (* *p* < 0.05). Moreover, HHA/BC was also significantly (# *p* < 0.05) efficacious in comparison to HHA + CS + cd for the tested metalloprotease negative modulation (Figure 5c). Cartilage matrix degradation is also connected to TIMP-2, a biomarker that has an important role in counteracting the activity of metalloproteases. The tested gels proved to significantly (* *p* < 0.05) increase the protein expression of this biomarker. In particular, TIMP-2 up-regulation was about 2.1 and 2.5-fold with respect to untreated cells in the presence of CS and BC, respectively. Moreover, the contribution to chondrocyte phenotype maintenance and production of specific cartilage components was measured through the assessment of COLII and AGN protein expression. Figure 5c shows that HHA + CS + cd and HHA/BC treated chondrocytes expressed a higher level of both COLII and AGN vs. pCTRL (* *p* < 0.05). Finally, HHA/BC significantly (* *p* < 0.05) prompted HAS-2 protein expression with respect to pCTRL (1.5-fold). In addition, in this case, the HHA/BC efficacy was superior to the HHA + CS + cd one (# *p* < 0.05).

### 2.6. Immunofluorescence Analyses

In our OA in vitro model, COLII protein expression was investigated using immunofluorescence in order to test the ability of HHA + CS + cd and HHA/CB to preserve chondrocyte-specific features counteracting the differentiation vs. fibroblastic phenotype. As shown in Figure 6, as expected, untreated pathological cells expressed a basal level of this biomarker, but it is interesting to note that in the presence of CS and BC, the green signal was more intense than pCTRL. In this context, a strong (* *p* < 0.05) intensity of COLII was found in HHA/BC-treated cells with respect to untreated pathological chondrocytes. Finally, HHA/BC proved a significant (# *p* < 0.05) effectiveness in specific phenotype maintenance with respect to HHA + CS + cd.

### 2.7. ELISA Assay

ELISA outcomes showed that the HA-based gels were effective in reducing IL-6 secretion. In detail, as shown in Figure 7, HHA/BC proved more efficacy in the down-regulation of this pro-inflammatory cytokine compared to HHA + CS + cd. In fact, in HHA/BC-based treated chondrocytes, IL-6 secretion was decreased in a significant way (* *p* < 0.05) with respect to the pathological control of about 22.9-fold. On the other hand, HHA + CS + cd was able to reduce IL-6 cellular secretion by about 5-fold.

## 3. Discussion

Despite the availability of several treatments based on anti-inflammatory drugs, food supplements (FS), and medical devices, the assessment of effective therapies for OA disease remains the principal objective for scientists and medical doctors [37]. Regarding medical devices, CE-marked intra-articular injective gels are mostly based on HA. As previously explained, the latter can be used linearly, as in natural biosynthesis, or chemically modified to improve stability and mechanical performance. Moreover, the combination of two different molecular weights HA recently established, namely hybrid stabilized cooperative complexes, may be employed [38]. However, some injective products contain not only HA but also other components, specifically GAGs, for example, the chondroitin sulfate [15,16,19].

To date, almost all the CS used either as food ingredients or as pharma-active molecules and as macromolecules in solution/combination with HA are of animal extractive origin [20,21,27,28,29]. In this respect, as previously discussed, the commercialized CS are very heterogeneous molecules. For instance, Stellavato and collaborators [19] and Pomin et al. [39] addressed the specific sulfation pattern related to bioactivity, considering keratan sulfate contamination of extractive CS. Additionally, the complete activation of biological pathways related to CS oral administration is not completely clear [39,40].

Considering the growing demand for vegan products and ethical commitment, religious constraints, and safety issues related to animal source products, the substitution of animal extractive CS with biotechnological or semisynthetic ones is foreseen. In this respect, Vessella et al. [41] reported a naturally identical molecule obtained using the semisynthetic approach, and a similar route was followed by Volpi and collaborators that claim shark-like chondroitin to be obtained and even approved by the Food and Drug Administration (FDA) as a food supplement [42]. In recent times, scientific investigations have suggested that CS, alone or in combination with other GAGs, may enhance chondrocyte metabolism, counteract the inflammatory process, and down-regulate the degradative enzymes [20,37]. In this context, the biological properties of BC have lately been tested in an OA animal model [43]. Thus, Cimini et al. [43] displayed that BC was as effective in reducing pain and affecting different specific biomarkers related to the inflammatory/degradation progression as CS in an OA mice model, highlighting the possibility of being able to replace extractive chondroitin with the biotechnological one. Furthermore, the recent outcomes on the bioactivity of biofermentative unsulfated chondroitin, when compared to CS, paved the way for its use in novel medical devices/oral formulations [43].

In this experimental setup, two commercial intra-articular devices were fully biophysically and biochemically characterized and compared. Certainly, the viscoelastic behavior is one of the key aspects to be investigated for these devices, responsible for mechanical stress absorption in the synovial joints during walking and running and also affecting ease of injection through thin needles (e.g., a gauge of 20–27) [44,45].

According to the mechanical spectra, HHA/BC and HHA + CS + cd samples showed similar rheological behavior in the experimental conditions tested, which are in line with other commercial products based on un-modified hyaluronic acid reported in the literature [46,47]. However, the slightly higher dynamic moduli and complex viscosity values registered for the HHA/BC hybrid complex could ensure a greater viscosupplementation capacity of the final product. Given the well-established capacity of reducing viscosity due to hydrogen hybrid cooperative complexes formation and stabilization, it can be argued that the MW of HHA in HHA/BC should be higher than the one used as raw materials in the preparation of HHA + CS + cd and/or possibly less affected/degraded by thermal sterilization.

Moreover, in this research, the activity of HHA/BC proved a superior resistance to enzymatic degradation than HHA + CS + cd; the weight fraction of molecules above 600 kDa was preserved 3-fold better. Thermal stabilization obtained following the patented protocols releases different interactions/entanglements between GAG chains; these are not obtained for HHA + CS + cd. In fact, hybrid complexes can shield HHA from the attack of BTH, as already evaluated in the formulation based on HHA and BC for dermal applications [48]. However, it has to be noticed that the enzymatic degradation is definitely slower for the chemically modified hyaluronan counterparts.

During OA progression, the hypothesis according to which the translocation of NF-kB from the cytoplasm to the nucleus triggers the gene and protein expression of markers responsible for inflammation and ECM degradation is well accepted [25,44,49]. For this reason, many studies have been focused on identifying molecules and/or active principles able to affect the expression of specific OA-related analytes (e.g., NF-kB, IL-6, TNF-α, and MMPs) and both chondroitin and hyaluronic acid were useful for this purpose [25,50,51].

Consistently with the scientific literature, in our pathological in vitro model, NF-kB-mediated inflammation was evident in untreated chondrocytes. In turn, the expression of this latter and related pro-inflammatory cytokines, IL-6 and TNF-α, were reduced via the treatments here tested [52]. Moreover, catabolic mediators, also leading to ECM degradation, like MMPs and aggrecans, resulted in a decrease [53]. Our data indicated that both samples were able to affect the expression of all these biomarkers; in particular, HHA/BC was more efficient than HHA + CS + cd in down-regulating NF-kB and IL-6. With the aim to improve the pattern of valuable markers and considering our previous studies (involving proteomic analyses), other specific analytes were also considered [32]. Specifically, the protein level of TIMP-2, which inhibits the MMPs action, and HAS-2, which contributes, through HA biosynthesis activation, to recovering the rheological features of a joint, resulted in enhancements.

In light of the obtained results, it is important to underline that BC and marine CS have similar features (e.g., MW) but also concrete differences, above all, the presence of sulfate groups that are considered directly correlated to their potential bioactivity [18,19]. In addition, keeping in mind these considerations and previous studies, the present outcomes again appear coherent, suggesting that the presence of sulfate groups may play a role in the activation of biological mechanisms [32,33].

Overall, the biochemical fingerprint here presented highlights that the biosynthetic activity of the pathological chondrocytes has been modulated towards a more physiological one via the GAGs-based treatments. In fact, on one side, there is the down-regulation of inflammatory and degradative molecules. On the other, in the extracellular matrix, structurally relevant macromolecules were prompted. Also, in these analyses, we could find a strict interconnection between the NF-kB inflammation pathway and chondrocyte viability and physiological functionality. Interestingly, the HHA/BC treatment and, with lower efficacy, the HHA + CS + cd, proved an impact at a molecular level in the OA in vitro model presented. HHA/BC can also retain the higher molecular weight fraction of hyaluronan longer, protecting it against enzymatic degradation. Sound performance of this injective product in vivo can be foreseen, possibly impacting pain and OA progression better than other HA-based injective medical devices.

## 4. Materials and Methods

### 4.1. Class III Medical Device Based on HA and CS or BC

Sinogel^®^ 3 mL (IBSA Farmaceutici, Lodi, Italy) (here referred to as HHA/BC) was kindly provided by IBSA Farmaceutici Italia. It contains hyaluronic acid sodium salt 2.4% (*w*/*v*) and unsulfated chondroitin 1.6% (*w*/*v*), and it represents a hybrid cooperative complex obtained via the patented thermal treatment for the stabilization of a high molecular weight and a low molecular weight (represented here by BC) hyaluronic acid fraction.

Dolatrox^®^ (Kolinpharma, Milan, Italy) (here referred to as HHA + CS + cd) was bought in a pharmacy; it is reported to contain hyaluronic acid sodium salt 2% (*w*/*v*), chondroitin sulfate 2% (*w*/*v*), and cyclodextrins 1% (*w*/*v*).

### 4.2. Rheological Characterization

Rheological characterization of the products was carried out using an Anton Paar Physica MCR301 oscillatory rheometer (Ostfildern–Scharnhausen, Germany), equipped with a cone-plate geometry CP50-2 (cone diameter 49.970 mm, truncation 207 μm, cone angle 1.995) and a Peltier system for temperature regulation. A preventive amplitude sweep test was performed at 1 Hz frequency over a strain amplitude range of 0.01–100% to derive the linear viscoelastic range (LVR) in which the moduli remained constant. Frequency sweep tests were then performed at a constant strain of 1% (within the LVR) over a frequency range of 0.1–10 Hz, and the values of the elastic modulus (G′), the viscous modulus (GG″), and the tan delta (GG″/G′) were measured [38]. Further, the viscoelastic moduli, tan δ, and the complex viscosity η* were extracted from the measurements at 0.5 Hz and 2.5 Hz, corresponding to the human walking and running frequency, respectively, and reported as mean ± SD.

### 4.3. Gels Susceptibility to Hyaluronidases

The Bovine testicular hyaluronidase (BTH, salt-free lyophilized powder with a specific activity of 890 U/mg, Sigma–Aldrich, Milan, Italy) catalyzed hydrolyses were accomplished as follows. Specifically, samples were diluted at 4 mg/mL in Dulbecco’s phosphate buffered saline (PBS without calcium and magnesium, Lonza Sales Ltd., Switzerland) in the presence of BTH (1 U/mL) and kept under stirring at 37 °C and 600 rpm. After 4 h of incubation, they were withdrawn and boiled for 10 min to inactivate the enzyme. Finally, they were filtered on 0.22 µm disposable membranes, diluted in deionized water, and analyzed using SEC-TDA to gain a full hydrodynamic characterization of the polymeric residual fractions. Each sample was treated at least in triplicate to check for repeatability, and hydrodynamic parameters obtained after the enzymatic hydrolysis were calculated as means ± SD. The degradation rate was determined as the decrease in sample molecular weight; specifically, the fraction (wt%) of macromolecules with Mw above 600 kDa was evaluated before and after the enzymatic action, and the residual percentage was consequently determined as:
(1)
$$∆{wt\%}_{Mw>600kDa}(\%)=\frac{{wt\%}_{Mw>600kDa}(post\;BTH)}{{wt\%}_{Mw>600kDa}(t0)}\times 100$$


A statistical study was performed on samples using a *t*-test analysis, and a *p*-value of 0.05 was chosen for significant differences.

### 4.4. OA In Vitro Model Setup

An in vitro model of OA was obtained following the experimental protocols well established in our laboratory [26,38]. Briefly, the knee cartilage samples, coming from OA-affected patients, were kindly provided by the Department of Medical and Surgical Specialties and Dentistry, University of Campania “Luigi Vanvitelli” (Naples, Italy). All the procedures were approved by the Internal Ethical Committee (AOU-SUN reg. no. 0003711/2015). To obtain a primary in vitro culture of pathological chondrocytes, cartilage tissues were enzymatically digested via a solution composed of collagenase type I at 3 mg/mL (*w*/*v*) and dispase at 4 mg/mL (*w*/*v*) at 37 °C on a shaking plate overnight. The next day, the cell suspension was filtered (70 μm, BD Falcon, Franklin Lakes, NJ, USA), centrifuged at 1500 rpm for 7 min (Eppendorf Centrifuge), washed with PBS, and re-centrifuged. The obtained chondrocyte pellet was re-suspended in DMEM, supplemented with fetal bovine serum (FBS) (10% *v*/*v*), penicillin–streptomycin (1% *v*/*v*), and amphotericin B (1% *v*/*v*). The specific phenotypic of cells was confirmed via fluorescence-activated cell sorting (FACS), as previously reported [26]. The cells used for the experiment presented here were at a second passage of in vitro culture.

All reagents for cell culturing were obtained by GIBCO (Thermo Fisher Scientific, Waltham, MA, USA).

### 4.5. Cellular Viability Assay

Primary chondrocytes were seeded in a 24-well standard plate (BD Falcon, Franklin Lakes, NJ, USA) (20 × 10^3^ cells/well) and incubated in the FBS-supplemented DMEM (untreated pathological cells, pCTRL) or the presence of the HHA/BC or HHA + CS + cd gels at final concentration of 4 mg/mL in the culture medium. The cells were treated for 72 h; within this experimental time, their viability was assessed using the Cell Counting Kit-8 (Dojindo EU GmbH) following the manufacturer’s protocol [54]. The optical densities of the supernatants were measured at 450 nm using a Beckman DU 640 spectrometer (Beckman, Milan, Italy) after specific treatments and incubation with a cell counting solution. The relative cell viability was calculated as a percentage of the maximal absorbance as follows:
(2)
$$Cell\;Viability\;vs\;pCTRL\;\left(\%\right)=\frac{mean\;OD\;treated\;cells}{mean\;OD\;untreated\;cells}\times 100$$


### 4.6. Gene Expression Analyses via qRT-PCR

After 72 h of treatments, the same cells used in the CCK-8 assay were harvested to isolate cellular RNA through TRIzol^®^ Reagent (Invitrogen, Milan, Italy) [35]. A quantitative real-time PCR was performed using the IQ™ SYBR^®^ Green Supermix (Bio-Rad Laboratories, Milan, Italy) using, for each sample, 1 µg of total RNA reversely transcribed into cDNA following the manufacturer’s protocol (Reverse Transcription System Kit, Promega, Milan, Italy). All the primer sequences used in this experimental setup are reported in Table 4. Each sample was analyzed in triplicate, and the glyceraldehyde-3-phosphate dehydrogenase (GAPDH) housekeeping gene was used to normalize the mRNA expression of specific analyzed genes. The variation of gene expression was calculated by normalizing the data about GAGs treated cells with respect to pCTRL, applying the Livak method (2^−ΔΔCt^) [55] and using Bio-Rad iQ™5 Optical System Software, Version 2.1 (Bio-Rad Laboratories, Milan, Italy).

### 4.7. Protein Analyses via Western Blotting

Primary chondrocytes were seeded in a 12-well standard plate (BD Falcon, Franklin Lakes, NJ, USA) (30 × 10^3^ cells/well) and in vitro cultivated in the presence or not (pCTRL) of GAGs-based gels for 7 days at a final concentration in the culture medium of 4 mg/mL. After one week, the expression of specific proteins related to the inflammatory process, ECM degradation/remodeling, and phenotype biomarker were analyzed using Western blotting (WB) following the previously described protocol [32,33]. In detail, the intracellular proteins were isolated from each sample using a radio-immunoprecipitation assay (RIPA buffer, Cell Signaling Technology) and quantified using the Bradford method [56]. In this context, 10 µg of protein was separated by an SDS-PAGE 10% polyacrylamide gel and transferred to a nitrocellulose membrane (GE, Amersham, UK). This latter was blocked with 5% non-fat milk in Tris-buffered saline and 0.05% Tween-20 (TTBS) for 30 min and incubated with primary antibodies against COMP-2 (Santa Cruz Biotechnology, Dallas, TX, USA used at a dilution of 1:500), NF-kB (Santa Cruz Biotechnology, Dallas, TX, USA used at a dilution of 1:250), MMP-13 (Santa Cruz Biotechnology, Dallas, TX, USA used at a dilution of 1:250), TIMP-2 (Santa Cruz Biotechnology, Dallas, TX, USA used at a dilution of 1:500), Collagen 2A2 (COLII) (Elabscience, Huston, TX, USA used at a dilution of 1:500), Aggrecan (AGN) (Santa Cruz Biotechnology, Dallas, TX, USA used at a dilution of 1:200), and HAS-2 (Santa Cruz Biotechnology, Dallas, TX, USA used at a dilution of 1:250) overnight at 4 °C. After that, TTBS was used to wash the membrane and dilute 1:10,000 the specific horseradish peroxidase-conjugated anti-mouse and anti-rabbit antibodies (Santa Cruz Biotechnology, Dallas, TX, USA). The secondary antibodies were incubated on the membrane for 1 h at room temperature. Each blot was developed via the ECL system (Elabscience, Huston, TX, USA), and an actin antibody (Santa Cruz Biotechnology, Dallas, TX, USA, diluted 1:1000) was used as a gel loading control. Lastly, the semi-quantitative protein expression analysis was accomplished through Image J software 1.8.0.

### 4.8. Immunofluorescence Analyses

To analyze the protein expression of COLII via immunofluorescence staining (IF), primary chondrocytes were seeded in a chamber slide (BD Falcon, Franklin Lakes, NJ, USA) (5 × 10^3^ cells/well) and in vitro cultivated in the presence or not (pCTRL) of GAG-based medical devices (4 mg/mL) for 24 h. After this time, supernatants were removed, and the samples were washed twice with PBS, fixed with paraformaldehyde 4% *w*/*v*, and permeabilized with a solution of Triton X-100 at 0.2% *v*/*v* in PBS. The primary antibody against COLII (Elabscience, Huston, TX, USA) (diluted 1:100) was incubated in each well overnight at 4 °C. Then, the slices were incubated with a FITC-conjugated goat anti-rabbit secondary antibody (Thermo Fisher Scientific, Waltham, MA, USA) (diluted 1:200) for 1 h at room temperature and covered using the mounting medium with DAPI-aqueous. The relative images were captured through an Axiovert 200 (Zeiss) fluorescence microscope and analyzed using AxioVision 4.8.2.

### 4.9. ELISA Assay

IL-6 secretion was quantified through ELISA assay (Boster Biological Technology Pleasanton, CA, USA). In detail, after 72 h of treatment, cellular supernatants were centrifuged in order to discard detached cells and/or debris (2000 rpm for 20 min at 4 °C). Each experiment was performed in triplicate, and the cytokine was quantified by a microplate reader (Bio-Rad Laboratories, Milan, Italy). The analytic concentrations were assayed via a standard curve according to the manufacturer’s instructions.

## 5. Conclusions

In this study, the biophysical, mechanical, and biological characteristics of two commercial gels based on hyaluronic acid and sulfated and unsulfated biofermentative chondroitins were assayed in an OA in vitro model based on human primary pathological chondrocytes. Our outcomes highlighted that although the GAGs in the tested gels present similar features, a few aspects may affect the final product’s properties. In particular, the origin of raw materials (e.g., extractive or biofermentative), the manufacturing process (e.g., purification steps), and the specific concentration. This set of factors was responsible for the gel’s biomechanical and biological activity, supporting the hypothesis that each formulation has definite biochemical and biological characteristics to be addressed to a specific need. For example, an HHA/BC sample, proving a superior beneficial effect on a few pathological joint biomarkers, may be proposed/selected for patients with advanced inflammation. In fact, the slower enzymatic degradation may also support the viscosupplementation longer, even in more damaged joints. Due to rheological similarities and a sound, despite the minor, beneficial effect of HA + CS + Cd, this injection may be suitable for early-stage OA patients. Finally, ethical and religious issues, general consumer awareness, and desirable sustainability approaches may support research, development, and use of medical devices from biotechnological processes rather than from extractive animal sources.

## Figures and Tables

**Figure 1 pharmaceuticals-16-01429-f001:**
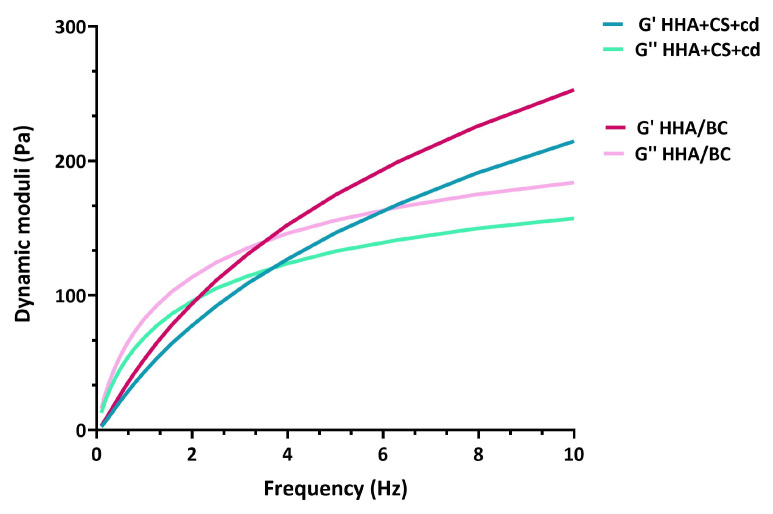
Mechanical spectra of the preparations as commercialized. HHA + CS + cd: hyaluronic acid sodium salt 2% (*w*/*v*), marine chondroitin sulfate 2% (*w*/*v*), and cyclodextrins 1% (*w*/*v*); HHA/BC: hyaluronic acid sodium salt 2.4% (*w*/*v*) and unsulfated chondroitin 1.6% (*w*/*v*).

**Figure 2 pharmaceuticals-16-01429-f002:**
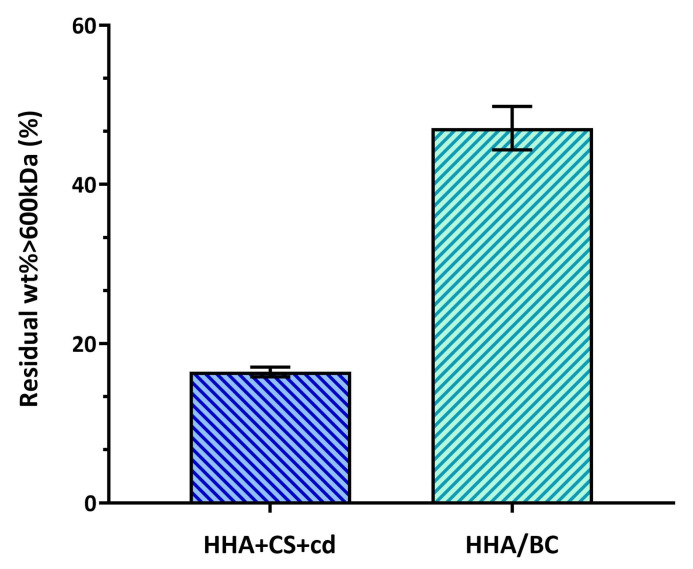
Weight fraction with molecular weight above 600 kDa residue after 4 h of incubation with BTH 1 U/mL.

**Figure 3 pharmaceuticals-16-01429-f003:**
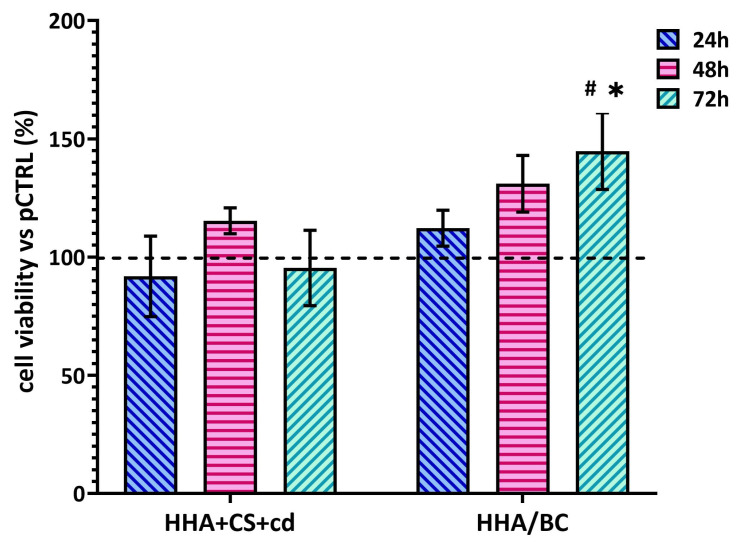
Cell viability assay was performed using CCK-8 staining. Data are presented as mean ± SD. * *p* < 0.05 *t*-test was used to compare the significance of each treatment with respect to pCTRL; # *p* < 0.05 vs. HHA + CS + cd.

**Figure 4 pharmaceuticals-16-01429-f004:**
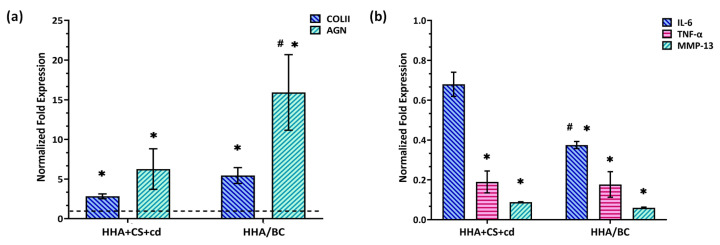
Expression analyses via qRT-PCR of (**a**) genes related to chondrocytes phenotype and cartilage structure (COLII and AGN) and (**b**) genes related to inflammation and cartilage degradation/remodeling (IL-6, TNF-α, MMP-13). Data are normalized with respect to pathological untreated cells (pCTRL). Results are presented as mean ± SD. * *p* < 0.05 *t*-test was used to compare the significance of each treatment with respect to pCTRL; # *p* < 0.05 vs. HHA + CS + cd.

**Figure 5 pharmaceuticals-16-01429-f005:**
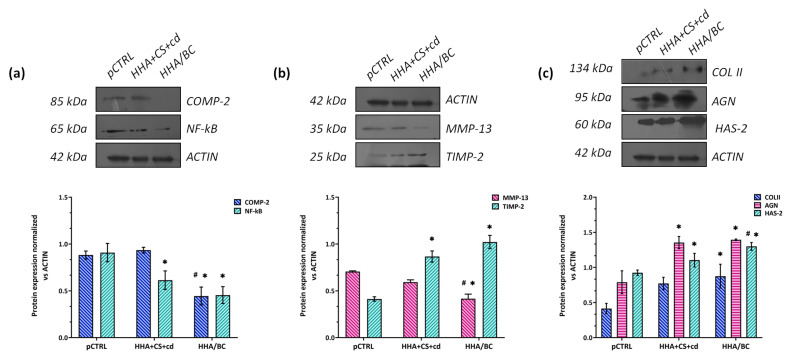
Expression levels evaluation via WB of proteins connected to (**a**) cartilage damage and inflammation (COMP-2 and NF-kB), (**b**) ECM degradation/remodeling (MMP-13 and TIMP-2), and (**c**) chondrocytes phenotype and cartilage structure (COLII, AGN, HAS-2). Densitometric analyses were performed, normalizing each protein expression with respect to ACTIN. * *p* < 0.05 *t*-test compared the significance of each treatment with respect to pCTRL. Outcomes are presented as the mean of two different WB ± SD. * *p* < 0.05 *t*-test was used to compare the significance of each treatment with respect to pCTRL; # *p* < 0.05 vs. HHA + CS + cd.

**Figure 6 pharmaceuticals-16-01429-f006:**
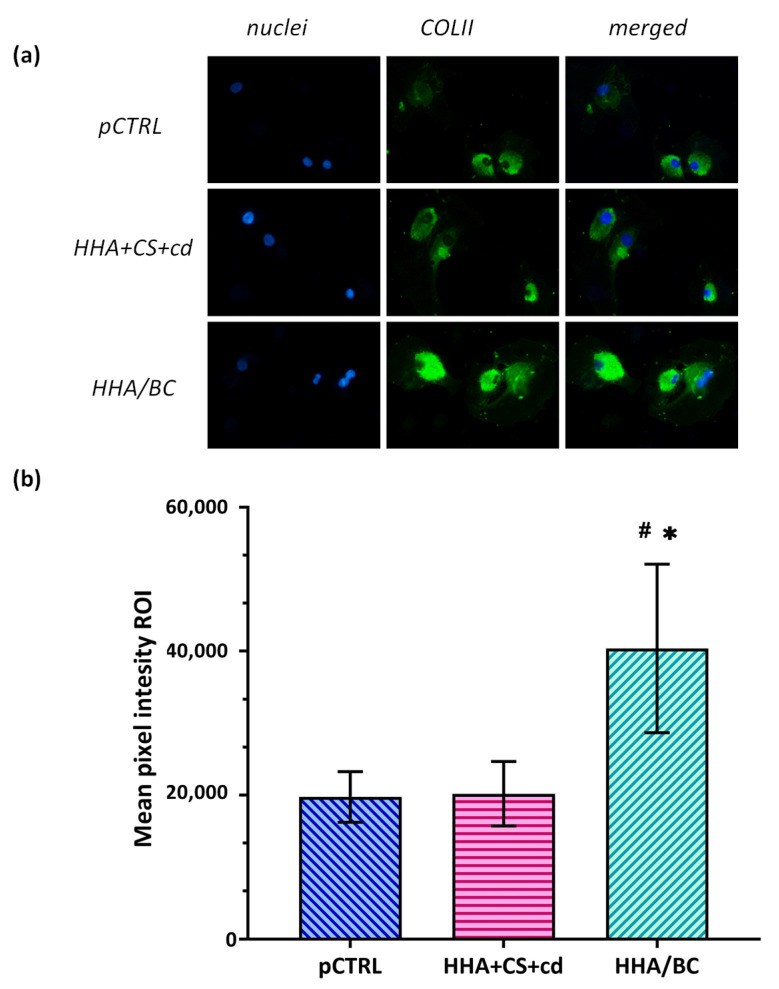
(**a**) Immunofluorescence staining of COLII in treated and untreated primary human chondrocytes. A FITC green antibody was used for collagen detection, while nuclei were stained in blue. Magnification 40×. (**b**) Graphs show the mean pixel intensity of COLII staining. * *p* < 0.05 *t*-test was used to compare the significance of each treatment with respect to pCTRL; # *p* < 0.05 vs. HHA + CS + cd.

**Figure 7 pharmaceuticals-16-01429-f007:**
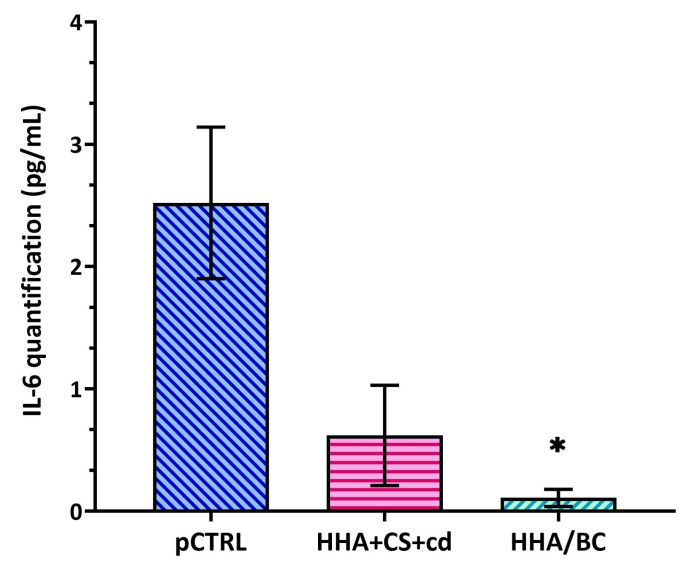
ELISA assay for IL-6 performed on human pathological chondrocytes treated for 72 h with HHA + CS + cd or HHA/BC in comparison to untreated cells (pCTRL). The values are presented as mean ± SD. * *p* < 0.05 *t*-test was used to compare the significance of each treatment with respect to pCTRL.

**Table 1 pharmaceuticals-16-01429-t001:** Dynamic moduli values at 0.5 Hz (walking frequency) and 2.5 Hz (running frequency) for samples as commercialized.

Sample	G′ (Pa)		GG″ (Pa)	
	0.5 Hz	2.5 Hz	0.5 Hz	2.5 Hz
HHA + CS + cd	21.6 ± 0.7	92.3 ± 1.6	45.7 ± 1.0	105.3 ± 3.2
HHA/BC	26.4 ± 2.0	111.4 ± 3.9	55.3 ± 5.0	124.7 ± 9.7

**Table 2 pharmaceuticals-16-01429-t002:** Tan δ at 0.5 Hz (walking frequency) and 2.5 Hz (running frequency) for samples as commercialized.

Sample	Tan δ	
	0.5 Hz	2.5 Hz
HHA + CS + cd	2.1 ± 0.1	1.1 ± 0.1
HHA/BC	2.2 ± 0.2	1.1 ± 0.1

**Table 3 pharmaceuticals-16-01429-t003:** Complex viscosity values at 0.5 Hz (walking frequency) and 2.5 Hz (running frequency) for samples as commercialized.

Sample	η* (Pa × s)	
	0.5 Hz	2.5 Hz
HHA + CS + cd	19.5 ± 0.7	16.0 ± 0.3
HHA/BC	37.1 ± 1.5	8.9 ± 0.2

**Table 4 pharmaceuticals-16-01429-t004:** Primer sequences used in qRT-PCR.

Gene Name	PCR Primer Sequence 5′→3′
*Glyceraldehyde-3-phosphate dehydrogenase (GAPDH)*	TGCACCACCAACTGCTTAGC GGCATGGACTGTGGTCATGAG
*Aggrecan (AGN)*	TCGAGGACAGCGAGGCC TCGAGGGTGTAGCGTGTAGAG
*Type II collagen (COLII)*	CAACACTGCCAACGTCCAGAT CTGCTTCGTCCAGATAGGCAA
*Matrix metallopeptidase 13* *(MMP-13)*	TCCCTGAAGGGAAGGAGC CTCGTCCAGGATGGCGTAG
*Interleukin-6 (IL-6)*	GTGGAGATTGTTGCCATCAACG CAGTGGATGCAGGGATGATGTTCTG
*Tumor necrosis factor alpha* *(TNF-α)*	CGAGTGACAAGCCTGTAGC GGTGTGGGTGAGGAGCACAT

## Data Availability

Data analyses by the authors are available upon reasonable request by contacting the corresponding author.

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
