# Peer review of "Hyaluronic Acid-Based Injective Medical Devices: In Vitro Characterization of Novel Formulations Containing Biofermentative Unsulfated Chondroitin or Extractive Sulfated One with Cyclodextrins"

_pharmaceuticals, 2023, doi:10.3390/ph16101429_

Round 1

Reviewer 1 Report

The paper is devoted to the study of commercial gels based on HA and CS, which are perspective for osteoarthritis treatment.

Comments:

1. The title of the paper is too long and confusing.

2. Abstract of the paper in my opinion should be more clear with regard to the statement of research objectives

3. The quality of all Figures is very poor and it is hard to analyze them properly cause the designations are barely visible.

4. In caption to Figure 1 the composition of the gels should be given for better understanding of results. The amount of CS and BC in wt% should be provided

5. The rheological properties (Page 9, lines 309-321) should be discussed taking into account the charge of chondroitin sulfate and BC. CS is highly charged molecule at all pHs, while BC should be more like HA. Kuhn's segment should be different for CS and BC, so elasticity of the chain is different. These facts should affect the mechanical properties of the gels. However, it is not so. The composition of the gels and ratios of components should be thus given and discussed.

6. Authors did not study and discuss the degradation properly. They compared HA/CS with HA/BS. Thus they have changed the structure of CS, but not of HA. Why different rate of digestion with hyaluronidase should be observed? Why authors didn’t used chondroitinase in this experiment? How CS and BC could affect the activity of hyaluronidase? I suggest authors to discuss these questions in the text of the paper.

7. Page 3, line 162. It is not explained by authors why one should expect toxicity for cells from natural components of extracellular matrix? Being natural products, these components could contain endotoxins, the content of which should be teste before any experiment with cells. Also, it is unclear what type of cells was used in this study and why authors have chosen this type of cells? The in formation should be given in the results part as well. In experimental part the source of primary cells should be provided. Animal or human?

8. The explanation of the observed experimental facts that changing of CS to BC result in different expression of the proteins should be given? This should be better discussed with application of literature.

9. In experimental part (Page 11, line 424) authors wrote: “The relative cell viability was calculated as a percentage of the maximal absorbance as following:”. But then on line 425 they name this parameter as “???????? (%)”. Usually, it is viability. Please unify. Line 421: the reference on “following the manufacturer’s protocol” should be provided. It is not clear why “optical densities of the obtained solutions” have been changed.

10. The statement in conclusions (Page 13, line 495) ”Our outcomes highlighted that, despite the GAGs present in the tested gels present similar features there are a lot of aspects that may affect the final products properties. In particular,  the origin of raw materials (e.g. extractive or biofermentative), the manufacturing process  (e.g. purification steps), and the specific concentration. This set of factors resulted responsible for the gels biomechanical and biological activity, supporting the hypothesis that each formulation has definite biochemical and biological characteristics to be addressed to a specific need.” is quite obvious and could be given without doing any experiments. More specific information based on the performed experiments should be given.

There are many mistakes, which make the reading challenging.

For example, on Page 10, line 331: "Our data indicated that both samples were able to affect the expression of all these biomarkers, in particular HHA/BC proved more effective than CS in down-regulating NF-kB and IL-6, in this in vitro model." (proved more effective?)

Page 9, line 281: "Moreover, the combination of two different molecular weights HA recently established, namely hybrid stablized cooperative complexes may be employed [29]."

These and many other mistakes seem to result not from bad knowlege of English, but from inaccuracy. Thus the text should be revised carefuly by the authors themselves.

Author Response

  1. The title of the paper is too long and confusing.

A: Thank you for your suggestion, the authors agree the proposed title was too long. Therefore “Comparative analyses of injective intra-articular medical devices based on hyaluronic acid:  in vitro evaluation of hybrid cooperative complexes containing biofermentative chondroitin and formulation with animal derived chondroitin sulfate and cyclodextrins” has been replaced by the following:

“Hyaluronic acid based injective medical devices: in vitro characterization of novel formulations containing biofermentative unsulfated chondrotin or extractive sulfated one with cyclodextrins”

  1. Abstract of the paper in my opinion should be more clear with regard to the statement of research objectives.

A:The abstract has been modified according to the referee suggestion.

  1. The quality of all Figures is very poor and it is hard to analyze them properly cause the designations are barely visible.

A:Thank you for the comment, probably there was a limited figures definition, whilts transferring the images to the journal format, however we decided to replace in the manuscript all figures with a better quality.

  1. In caption to Figure 1 the composition of the gels should be given for better understanding of results. The amount of CS and BC in wt% should be provided.

A:Figure 1 caption was modified according to thesuggestion.

  1. The rheological properties (Page 9, lines 309-321) should be discussed taking into account the charge of chondroitin sulfate and BC. CS is highly charged molecule at all pHs, while BC should be more like HA. Kuhn's segment should be different for CS and BC, so elasticity of the chain is different. These facts should affect the mechanical properties of the gels. However, it is not so. The composition of the gels and ratios of components should be thus given and discussed.

A:We agree with the referee that the polyanionic nature of CS is different from BC and also HA. However, as the referee can see here we are using a high MW Hyaluronan that is in fact the preminent compound in terms of viscoelastic behaviour. The topic suggested by the reviewer is indeed interesting and in our opinion can be deepened in further studies where similar MW HA (e.g. ~50 KDa ) can be compared to the CS and BC with a specific hydrodynamic and rheological analyses. However, we believe that being BC and CS only part of the formulation, these information, related to a theoretical model,  are not very relevant to the aim of this paper.

  1. Authors did not study and discuss the degradation properly. They compared HA/CS with HA/BS. Thus they have changed the structure of CS, but not of HA. Why different rate of digestion with hyaluronidase should be observed? Why authors didn’t used chondroitinase in this experiment? How CS and BC could affect the activity of hyaluronidase? I suggest authors to discuss these questions in the text of the paper.

A:Thank you for the comment, we improved the discussion according to your suggestion.

Regarding your observation, the authors have to highlight that a previous study [D’Agostino 2022, HHA/BC] demonstrated the potential of HHA/BC hybrid cooperative complex in terms of reducing viscosity and increasing injectability, while maintaining a high concentration of GAG in solution, and especially in slowing down the degradation process of the formulation in the presence of hyaluronidases (with respect to comparable HHA). In the framework of this experimental research HHA/BC proved a superior resistance to enzymatic degradation than HHA+CS+cd, measured as the residual high molecular weight fraction of Hyaluronic acid, that is the relevant counterpart for viscoelastic properties, very important for the foreseen application.   . Thermal stabilization obtained following the patented protocols release different interaction/entanglements between GAGs chains; there is no evidence reported to the best of our knowledge that the 3 components mixture can resemble the inventive step previously recalled.  In fact, hybrid complexes can shield HHA from the attack to BTH as already evaluated on the formulation based on HHA and BC for dermal applications.  However, it has to be noticed that the enzymatic degradation is definitely slower for the chemically modified hyaluronan counterparts.

The Chondroitinase ABC that is known to have a certain hyaluronidase activity  is derived by Proteus mirabilis, here the enzyme BTH was chosen because it is the most frequently used to assess comparative stability between diverse HA based formulations in the literature, being also the one with better homology to human corresponding protein.

The discussion was modified as following:

Page 12, lines 368-375: “Moreover, in this research activity HHA/BC proved a superior resistance to enzymatic degradation than HHA+CS+cd. In fact, the weight fraction of molecules above 600 kDa was preserved 3-fold better. Thermal stabilization obtained following the patented protocols release different interaction/entanglements between GAGs chains; these are not obtained for HHA+CS+cd. In fact, hybrid complexes can shield HHA from the attack to BTH as already evaluated on the formulation based on HHA and BC for dermal applications [48].  However, it has to be noticed that the enzymatic degradation is definitely slower for the chemically modified hyaluronan counterparts”.

  1. Page 3, line 162. It is not explained by authors why one should expect toxicity for cells from natural components of extracellular matrix? Being natural products, these components could contain endotoxins, the content of which should be teste before any experiment with cells. Also, it is unclear what type of cells was used in this study and why authors have chosen this type of cells? The information should be given in the results part as well. In experimental part the source of primary cells should be provided. Animal or human?

A: We agree with the referee the point could be confusing to the readers.

As explained in the text, we used registered medical devices, therefore these products have already passed all the requires safety and toxicity evaluation needed before notified body acceptance and CE certification. These kind of trials are not using pathological models, in our experimental set up, the stressed cells coming from very damaged joints may be more susceptible to  eventually occurring contaminants even at very low concentration. We tried to clarify the point , correcting the text as following:  

Page 5, lines 190-195: “Although the two products here tested are registered in Europe, their effect in a pathological in vitro model, based on chondrocytes, is interesting to highlight eventually occurring effects due to contaminants even at very low concentration (e.g., trace elements, metals, endotoxins, etc.). Therefore, our data, that showed human pathological chondrocyte viability under treatments are relevant”.

Moreover, we explained within the end of introduction that the employed cells were “human pathological chondrocytes” (Page 3, line 129).

In particular, this kind of cells was selected because the model has been well accepted in the scientific literature. In fact, several scientific reports described the use of in vitro models of osteoarthritis (OA) pathology, obtained insulting chondrocytes with IL-1β or IL17 in order to mimic the OA catabolic and inflammatory process [1-2]. Whereas OA in vitro model established in our laboratories [3] was based on articular cells isolated from OA affected patients already presenting an ongoing inflammatory/degradative process. In this respect, the study here reported may better resembling actual/real in vivo conditions and should be useful to shade light on the biochemical and cellular mechanisms beyond thus, tissue repair.

  1. Calamia V. et al. “Pharmacoproteomic study of three different chondroitin sulfate compounds on intracellular and extracellular human chondrocyte proteomes,” Molecular & Cellular Proteomics, vol. 11, no. 6, Article ID 013417, p. M111, 2012.
  2. Lourido L. et al. “Secretome analysis of human articular chondrocytes unravels catabolic effects of nicotine on the joint,” Proteomics - Clinical Applications, vol. 10, no. 6, pp. 671–680, 2016.
  3. Vassallo V et al. “Unsulfated biotechnological chondroitin by itself as well as in combination with high molecular weight hyaluronan improves the inflammation profile in osteoarthritis in vitro model”. J Cell Biochem. 2021 May 31;122(9):1021–36. doi: 10.1002/jcb.29907.

Finally, the required information has been inserted in the text.

Page 5, lines 188-190: “The in vitro model here proposed is based on primary chondrocytes isolated from OA affected patients thus presenting an ongoing inflammatory/degradative process. In this way, the experimental set-up may better resemble in vivo conditions.”

  1. The explanation of the observed experimental facts that changing of CS to BC result in different expression of the proteins should be given? This should be better discussed with application of literature.

A:Thank you for the comment, we modified the discussion according to the referee suggestion and inserted new references specifically relevant to the topic in order to  better clarify our proposed results analyses to the readers.  

Page 11, lines 331-337: “Up to date, almost all the CS used either as food ingredients or as pharma active molecules, and as macromolecules in solution/combination with HA are of animal extractive origin [20,21,27-29]. In this respect, as previous discussed, the commercialized CS are very heterogenous molecules. For instance, Stellavato and collaborators [19], and Pomin et al. [39], addressed the specific sulfation pattern also related to bioactivity, considering keratan sulfate contamination of extractive CS.  Besides, the complete biological pathways activation related to CS oral administration are not completely clear [39,40].”

 Page 12, lines 376-382: “During OA progression, it is well accepted the hypothesis according to which the translocation of NF-kB from the cytoplasm to the nucleus triggers the gene and protein expression of markers responsible for inflammation and ECM degradation [25,49,50]. For this reason, many studies have been focused on the identification of molecules and/or active principle able to affect the expression of specific OA related analytes (e.g., NF-kB, IL-6, TNF-α, MMPs) and both chondroitin and hyaluronic acid resulted useful for this purpose [25,51,52]”.

Page 12, lines 383-385: “Consistently with scientific literatures, in our pathological in vitro model, NF-kB mediated inflammation was evident in untreated chondrocytes. In turn, the expression of this latter and related pro-inflammatory cytokines IL-6 and TNF-α, were reduced by the treatments here tested [53]”.

Page 12, lines 394-399: “In light of the obtained results, it is important to underline that BC and marine CS, have similar features (e.g., MW) but also concrete differences, above all the presence of sulfate groups, that is considered directly correlated to their potential bioactivity [18-19]. Keeping in mind these considerations and previous studies, the present outcomes appear again coherent, suggesting that the presence of sulphate groups may play a role in bio-logical mechanisms activation [32,33]”.

  1. In experimental part (Page 11, line 424) authors wrote: “The relative cell viability was calculated as a percentage of the maximal absorbance as following:”. But then on line 425 they name this parameter as “???????? (%)”. Usually, it is viability. Please unify. Line 421: the reference on “following the manufacturer’s protocol” should be provided. It is not clear why “optical densities of the obtained solutions” have been changed.

A:We modified according to the reviewer suggestion as following:

Page 14, lines 486-487: “The relative cell viability was calculated as a percentage of the maximal absorbance as following: Cell Viability vs pCTRL (%) = (mean OD treated cells) / (mean OD untreated cells) ×100”.

The relative reference has been inserted: “Dojindo Cell Counting Kit-8Handbook https://goldbio.com/uploads/documents/429b0ad1486d8e41a337c3f3c2d3f44d.pdf”.

As explained in the user manual there is a change of color intensity of supernatants directly correlated to cell viability. In our context, “optical densities of the obtained solutions” means the obtained supernatants of a specific intensity color depending on treatments. However, we clarified this point within the text as following:

Page 14, lines 482-484: “The optical densities of the supernatants were measured at 450 nm using a Beckman DU 640 spectrometer (Beckman, Milano, Italy) after specific treatments and incubation with Cell Counting solution.”

  1. The statement in conclusions (Page 13, line 495) ”Our outcomes highlighted that, despite the GAGs present in the tested gels present similar features there are a lot of aspects that may affect the final products properties. In particular, the origin of raw materials (e.g. extractive or biofermentative), the manufacturing process (e.g. purification steps), and the specific concentration. This set of factors resulted responsible for the gels biomechanical and biological activity, supporting the hypothesis that each formulation has definite biochemical and biological characteristics to be addressed to a specific need.” is quite obvious and could be given without doing any experiments. More specific information based on the performed experiments should be given.

A: Considering the referee  comment, we attempted at improving the conclusion adding the following sentence.

Page 16 lines 584-592: “For example, HHA/BC sample, proving a superior beneficial effect on few pathological joint biomarkers may be proposed/selected for patients with advanced inflammation. In fact, also the slower enzymatic degradation may support the viscosupplementation longer even in more damaged joints. Due to rheological similarities, and a sound, despite minor, beneficial effect of HA+CS+Cd, this injection may be suitable for early-stage OA patients. Finally, ethical, and religious issues, a general consumer awareness and desirable sustainability approaches may support research, development, and use of medical devices from biotechnological processes rather than from extractive animal sources.

Comments on the Quality of English Language

There are many mistakes, which make the reading challenging.

For example, on Page 10, line 331: "Our data indicated that both samples were able to affect the  expression of all these biomarkers, in particular HHA/BC proved more effective than CS in down-regulating NF-kB and IL-6, in this in vitro model." (proved more effective?)

A We corrected as following:

Pag 12, lines 388-389: “was more efficient”.

Page 9, line 281: "Moreover, the combination of two different molecular weights HA recently established, namely hybrid stablized cooperative complexes may be employed [29]."

We read but it was written “stabilized”. However, we checked all manuscript with more attention.

These and many other mistakes seem to result not from bad knowlege of English, but from inaccuracy. Thus the text should be revised carefuly by the authors themselves.

A: Thank you, we revised the text very carefully.

Reviewer 2 Report

This paper presents a comparative analysis of injectable articular medical devices based on hyaluronic acid: in vitro evaluation of complex compounds containing bio-fermented chondroitin and animal-derived chondroitin sulfate and cyclodextrin preparations. Overall, it is an interesting article, but a major revision is needed before it can be considered for publication.

Comment:

1. There are many abbreviations in the article, so it is suggested that the author add a list of abbreviations, which is convenient for readers to read and only the phrases that occur repeatedly in the article need to be abbreviated.

2. The introduction is confusing to read. I suggest that the author check the logic of the introduction.

3. The research on osteoarthritis has been included in “Yang L, Sun L, Zhang H, et al. Ice-inspired lubricated drug delivery particles from microfluidic electrospray for osteoarthritis treatment[J]. ACS nano, 2021, 15(12): 20600-20606.” and “Lei, Q. Zhang, G. Kuang, X. Wang, Q. Fan, F. Ye, Functional biomaterials for osteoarthritis treatment: From research to application. Smart Med. 2022, e20220014.” The authors are suggested to discuss them or make a comparison with them.

4. The quality of the pictures in the article is very poor, I can't see clearly the contents of the pictures and can't make a judgment on them.

5. The Western blotting results are placed too casually. The strips should be placed neatly near the middle line, and the internal parameters should be placed uniformly in the last row.

6. The Gratings of the text are not uniform, why the text after 4.1 is not placed at the top?

Minor editing of English language required

Author Response

  1. There are many abbreviations in the article, so it is suggested that the author add a list of abbreviations, which is convenient for readers to read and only the phrases that occur repeatedly in the article need to be abbreviated.

A: Thank you for the comment, as suggest we provided a list of abbreviation.

  1. The introduction is confusing to read. I suggest that the author check the logic of the introduction.

A: We checked the introduction and improved according to the referee suggestion. In addition, we inserted in the opportune context the references required at point 3.

  1. The research on osteoarthritis has been included in “Yang L, Sun L, Zhang H, et al. Ice-inspired lubricated drug delivery particles from microfluidic electrospray for osteoarthritis treatment[J]. ACS nano, 2021, 15(12): 20600-20606.” and “Lei, Q. Zhang, G. Kuang, X. Wang, Q. Fan, F. Ye, Functional biomaterials for osteoarthritis treatment: From research to application. Smart Med. 2022, e20220014.” The authors are suggested to discuss them or make a comparison with them.

A: As suggested, we inserted the specific references in the manuscript, since we agree with the referee that they are relevant.

  1. The quality of the pictures in the article is very poor, I can't see clearly the contents of the pictures and can't make a judgment on them.

A: Thank you for the comment, as explained to Reviewer #1, we re-inserted in the manuscript all figures with a better quality.

  1. The Western blotting results are placed too casually. The strips should be placed neatly near the middle line, and the internal parameters should be placed uniformly in the last row.

A: Considering your suggestion, we modified the figure trying to improve the quality.

However, the biomarker are places with respect to their electrophoretic migration (i.e., MW), the sequence was always repeating, namely 1st pCTRL-2nd HHA+CS+cdr and finally 3rd HHA/BC. In the attempt to make easier to the reader to read the results.  

  1. The Gratings of the text are not uniform, why the text after 4.1 is not placed at the top?

A: We uniformed the gratings, as requested.

Comments on the Quality of English Language

Minor editing of English language required

A: Thank you, as suggested also by the Reviewer #1 we revised the text very carefully.

Reviewer 3 Report

This work conducted a throughout study on the issue of injective intra-articular medical device. A big problem is that figures are all vague; however, the results and the corresponding discussion are all in a correct form. Thereby, I recommend it a major revision. The detailed comments are tabulated as follows.

1. A more concise title should be organized. The presented title of this work is somehow obscure, which is quite hard for readers to catch their intention.

2. In general, all the figures in the context own the poor resolution; I cannot clearly distinguish the marked text in the figure. Sometimes, I must guess the expression of the meanings of each figure. Thus, please prepare the clearer figures in the revised version.

3. A comprehensive background survey should be conducted on the Introduction section, especially for the chemicals used herein; For instance, the wide application of cyclodextrin and its derivatives (e.g., Corros. Sci., 2023, 212: 110957). Others such as chondroitin sulfate should also be done for a strong background survey.

4. For the rheological property given in Figure 1, the variation difference of responses at low-frequency region from that at high-frequency region should be discussed. In addition, the point-line curves are common for the facile distinguishing the characteristic modulus at a certain frequency.

Author Response

Comments:

  1. A more concise title should be organized. The presented title of this work is somehow obscure, which is quite hard for readers to catch their intention.

A: Thank you for your suggestion, as explained to Reviewer #1, the authors agree the title was too long. In this respect the first title: “Comparative analyses of injective intra-articular medical devices based on hyaluronic acid: in vitro evaluation of hybrid cooperative complexes containing biofermentative chondroitin and formulation with animal derived chondroitin sulfate and cyclodextrins” has been replaced by the following:

Hyaluronic acid based injective medical devices: in vitro characterization of novel formulations containing biofermentative unsulfated chondrotin or extractive sulfated one with cyclodextrins.

  1. In general, all the figures in the context own the poor resolution; I cannot clearly distinguish the marked text in the figure. Sometimes, I must guess the expression of the meanings of each figure. Thus, please prepare the clearer figures in the revised version.

A: We provided all figures with a better quality.

  1. A comprehensive background survey should be conducted on the Introduction section, especially for the chemicals used herein; For instance, the wide application of cyclodextrin and its derivatives (e.g., Corros. Sci., 2023, 212: 110957). Others such as chondroitin sulfate should also be done for a strong background survey. 

A: Thank you, as also suggested by Reviewer #2, we modified the introduction according to your suggestion and inserted the specified references.

  1. For the rheological property given in Figure 1, the variation difference of responses at low-frequency region from that at high-frequency region should be discussed. In addition, the point-line curves are common for the facile distinguishing the characteristic modulus at a certain frequency.

A: The rheological profiles obtained for both commercial formulations resemble those of linear HA based formulations, which are largely dependent on the strength of the HA chains network in solution. [1].

Mechanical spectra and relative dynamic moduli highlighted the typical viscoelastic behavior of entangled network, that turns from viscous to elastic with increasing frequency.

At low frequencies, in fact the polymer chains have enough time to disentangle, and the behavior is similar to viscous liquids (the loss modulus dominates); whereas at higher frequencies, i.e., short timescale, the polymer chains do not manage to escape the entanglements and act as an elastic network (the storage modulus dominates) [2].

The crossover point represents the frequency at which the polymer starts to exhibit a solid-like behavior; the latter is usually showed by crosslinked hyaluronan based products over a wider range of frequencies.

  1. Krause WE, Bellomo EG, Colby RH. Rheology of sodium hyaluronate under physiological conditions. Biomacromolecules. 2001 Spring;2(1):65-9. doi: 10.1021/bm0055798. PMID: 11749156.
  2. Dodero, Andrea et al. “A micro-rheological and rheological study of biopolymers solutions: Hyaluronic acid.” Carbohydrate polymers vol. 203 (2019): 349-355. doi:10.1016/j.carbpol.2018.09.072.

Round 2

Reviewer 1 Report

Authors have met all of my comments and really improved their manuscript.

English is Ok

Reviewer 2 Report

accept

Minor editing of English language required

Reviewer 3 Report

After re-evaluation of this manuscript, I recommend it acceptance in the present form.